# Knowledge, Implementation, and Gaps of Gender-Based Violence Management Guidelines among Health Care Workers

**DOI:** 10.3390/ijerph20075409

**Published:** 2023-04-05

**Authors:** Caroline Mtaita, Elvis Safary, Katanta Simwanza, Rose Mpembeni, Samuel Likindikoki, Albrecht Jahn

**Affiliations:** 1Heidelberg Institute of Global Health, Im Neunheimer Feld 130/3, 69120 Heidelberg, Germany; 2EngenderHealth Tanzania, Mwai Kibaki Road, 113 Mikocheni, Dar es Salaam 78167, Tanzania; 3School of Public Health and Epidemiology, Muhimbili University of Health and Allied Sciences, Dar es Salaam 65105, Tanzania; 4School of Medicine, Muhimbili University of Health and Allied Sciences, Dar es Salaam 65001, Tanzania

**Keywords:** gender-based violence, AGYW, knowledge, health care workers, low-income countries, Sub-Saharan Africa

## Abstract

(1) Background: Gender-based violence (GBV) is widespread globally and has a myriad of adverse effects but is vastly under-reported. Health care workers are among the first responders in GBV. The objective of this study was to assess the knowledge of health workers with regard to GBV and related management guidelines and implementation. (2) Methods: The study employed a descriptive, sequential mix-method study, beginning with the quantitative part, followed by the qualitative component. Qualitative analysis was conducted using a content framework approach. (3) Results: More than two-thirds (71.79%) of health workers were found to be generally knowledgeable about gender-based violence; however, only 36.9% had good knowledge about gender-based violence management guidelines for gender-based violence and the mean value for all the items was less than 3 which indicates poor knowledge of the management guideline. Additionally, only 36.8% found the gender-based violence management guidelines useful and practical in clinical care for gender-based violence cases. (4) Conclusions: The finding of this study revealed that knowledge of gender-based violence management guideline was not adequate among health workers and rarely used during management of GBV cases. This calls for continuous training and specific refresher courses, including on-site practical sessions, professionals’ mentorship, and supervision.

## 1. Introduction

Gender-based violence (GBV) among adolescent girls and young women (AGYW) is an imperative public health concern that draws the attention of a wide spectrum of professional health care workers (HCW). GBV affects women’s wellbeing with adverse health outcomes [1] consequently leading women who are abused to extensively seek health services [2].

Health systems have a crucial role in a multisector response to strengthen the role of HCWs in responding to violence against women [3]. Most women come into contact with health care settings; this makes the health care setting an important and safe place where women can confidently disclose experiences of violence and receive supportive response [4].

In Latin American countries, women who had been victims of violence indicated that an increase in the severity and frequency of the violence motivated them to seek health care [5]. Additionally, some reported that they had received good care and perceived the care to be good knowledge among health workers. However, several factors were also identified to hinder women from accessing health services; these include preference of settlement within the family and community, lack of awareness of services, economic considerations since most of them expressed concern about their ability to support themselves and also feeling of guilt, fear, and self-blame [6]. In Serbia, one in four women (23.7%) aged 15–49 years reported violence at least once in their life [7]. In another study conducted in Serbia to identify whom women approach for help in case of violence, 22% of women sought health care from a formal institution. Most women reported the highest satisfaction with services from health centers and legal advice and the lowest for police services. This report cited higher satisfaction with higher knowledge of GBV and good implementation of guidelines. Women who did not visit cited a lack of trust in formal institutions and fear of undesirable consequences [8].

In some African countries, such as Zimbabwe, two-thirds of women aged 18–24 years have experienced some form of violence. Another study in Zimbabwe reported that 32% of women experienced violence [9]. Strikingly, women who experienced physical or sexual assault use health services more frequently than the non-abused peers citing satisfaction with how health workers handled them, which is attributed to a better knowledge of GBV [10]. In Kenya, GBV is recognized as a significant problem, particularly in the rural areas and “informal” settlements of Kenya [11]. According to the 2014 Kenya Demographic Health survey, 14% of women reported being abused, yet only a third reported seeking help. However, there were concerns about underreporting of GBV cases, particularly in seeking health services.

To implement the GBV program effectively and to combat violence among AGYW, it is necessary for health professionals to have adequate knowledge of and comply with the implementation of GBV-related clinical guidelines. Their implementation depends on how much knowledge they have and how often they update their knowledge. A scoping review on sexual and GBV in East Africa reported that there is willingness by health care providers to care for GBV survivors, but this is hampered by concerns regarding the ability to provide care due to resource constraints [12].

In the Tanzanian context, violence is considered gendered because it is violence associated with women. Moreover, GBV has a greater impact on women and girls since they are most often survivors and suffer greater physical damage [13]. It results from gender norms, gender discrimination, and social and economic inequalities [13]. According to the Tanzania Demographic Health Survey (TDHS) 2015–2016, the percentage of women who access health services following GBV is still very small (1.1%), despite the Government’s effort to prevent and respond to GBV specifically on health response by increasing GBV health services in health facilities. Strategies put in place in line with GBV health response include the provision of GBV management guidelines for all health facility levels at all health facility entry points, training of HCWs, GBV community outreach programs, the establishment of one-stop centers [14], establishment of a national recording system for GBV cases, harmonizing of GBV indicators into the National Health Management Information System (HMIS) and the District Health Information System (DHIS) and assigning GBV focal person for each health facility [14].

Health professionals are in an exceptional position to identify the problem, contribute to its prevention, and assist gender-based violence survivors. This is because health facilities are probably one of the few public institutions that most girls and women interact with at some point in their lives for example during pregnancy and deliver care, family planning, pediatric care, and general health need [15]. Alazmy et al. 2011 [16], in their paper on gender differences in knowledge and attitude of primary health care staff towards violence, reported that cultural and traditional norms in community care have an impact on gender equity. Gender differences in knowledge can thus have an effect during the implementation of GBV guidelines.

The absence of screening for GBV among HCWs could be regarded not only as a reflection of patriarchy in the health systems but also as a form of “institutional silence” on GBV [17]. Key lessons from several African countries have emerged from experience dealing with violence against women within the health sector. Interventions must go beyond training and curricula reform and utilize a system-wide approach, including changes in policies, procedures, and attention to privacy and confidentiality. Franzoi et al. (2011) [18] in their paper described the need to broaden professional education to enable workers to deal with gender-based violence through the effective implementation of guidelines to address violence against women.

In health care settings where providers are well-trained, caring, and sensitive, most women respond positively to being asked about their exposure to violence [19,20,21]. Several studies have focused on violence among AGYW as well as feedback from health workers regarding violence. However, since the development of the Tanzanian GBV management guideline in 2010, to our knowledge, no studies have evaluated health care workers’ knowledge of GBV and implementation of GBV management guidelines among HCWs. The main objective of this study was to assess the knowledge, implementation, and gaps of GBV management guidelines among HCWs. The outcome of the findings is expected to help in the improvement of policies on the provision of GBV health services among HCWs.

## 2. Materials and Methods

### 2.1. Study Setting

The study sites were in the Temeke and Kinondoni districts of Dar es Salaam. Dar es Salaam is the largest city in Tanzania and comprises five districts (Temeke, Kinondoni, Ilala, Ubungo, and Kigamboni). Temeke and Kinondoni districts were selected for this study because AGYW have been reported by McCleary-Sills, J. et al. (2013) to have the highest reported cases of GBV in Dar es Salaam [22].

### 2.2. Study Design and Sample Size

This was a facility-based study that analyzed both quantitative and qualitative data obtained from HCWs (Medical Officers, Assistant Medical officers, Clinical Officers, Registered Nurses, and Enrolled Nurses) from Temeke and Kinondoni districts in Dar es Salaam, Tanzania. The explanatory sequential mixed methods design was used [23]. The research started with a quantitative part where the investigators assessed knowledge and implementation of GBV management guidelines among HCWs. Findings from the quantitative section informed the design of a qualitative follow-up, which aimed at gathering experiential information from HCWs regarding their perspective on knowledge, implementation, and gaps in the Tanzanian GBV management guideline. The step-by-step mixed methods methodology employed in conducting the study is shown in Figure 1.

### 2.3. Data Collection and Sampling Process

The multi-stage sampling procedure was employed to sample HCWs from health facilities in Kinondoni and Temeke. A list of all Government health facilities was obtained from the Tanzanian National Registry of Health facilities. Health facilities were further grouped into Dispensaries, Health Centers, and Hospitals. All HCWs at the outpatient department (OPD), care and treatment center (CTC), and reproductive and child health (RCH) were involved in the study. Twenty (20) HCWs were purposively selected for the qualitative part of the study; 10 HCWs from each district were included in the study. Quantitative data were collected using a structured questionnaire adapted from versions of instruments used in other studies based on an extensive review of the literature [24,25]. This had both open-and-close-ended questions related to various aspects of GBV. HCWs were approached and requested to participate; only those who assented and consented were included in the study.

### 2.4. Data Analysis

Researchers were involved in the preliminary data analysis for arising themes. Recorded data were translated from Swahili to English and then organized in an excel file for ease of retrieval. The researcher repeatedly read all the scripts for familiarization and deep understanding of the information. The thematic framework was drawn from the research objectives and issues that arose from the interview. The data were indexed and grouped into themes according to level of generality for easy retrieval, review, and further exploration. Quantitative data were collected and compiled using Microsoft Excel. Descriptive statistics included frequencies of each variable. Likert scale mean scoring: knowledge was determined based on the argument that a mean score of 3 on the Likert scale represents no opinion, mean score of less than 3 represents poor knowledge, and greater than 3 represents good knowledge [26]. The range of interpreting the Likert scale mean score was given as follows: 1.0–2.4 (poor knowledge), 2.5–3.4 (no opinion), and 3.5–5.0 (good knowledge). For analysis, we employed STATA 13.0 statistical software.

### 2.5. Ethical Considerations

The study was granted ethical approval by the Medical Research Coordinating Committee (MRCC) of the National Institute for Medical Research (NIMR) in Tanzania (NIMR/HQ/R.8a/Vol.IX/2986) and the Ethics Committee of the Medical Faculty of Heidelberg University (S-737/2018). Approval to work in the study wards was obtained through official permission from respective central and local government authorities and leaders. Permission to access the HCWs was granted by the Hospital Administrator. All participants provided written informed consent for participation in the study. Confidentiality will be maintained for adolescent girls, and young women as no names or identification of either the participants was used in this study.

## 3. Results

We present our results according to three topic areas: (1) knowledge of the definition of gender-based violence among HCWs, (2) knowledge of GBV management guidelines for health sector response for prevention of GBV among HCW, (3) implementation of a guideline for the health sector response to and prevention of GBV, and (4) perceived gaps in the GBV management guideline: HCWs perspectives.

### 3.1. Socio-Demographic Characteristics

Table 1 illustrates the socio-demographic characteristics of our study participants. Out of 202 HCW, 138 (68.3%) were from Kinondoni district. The majority of HCW were in the age group 34–44 years (43.6%; mean age = 39; SD = 9.45). Most of the HCW were females (71.3%). Regarding the level of education, the majority had a diploma (86.4%), and most (60.4%) worked at health centers. The majority of HCW were nurses/midwives (60.4%), and most worked in outpatient department (31.5%).

### 3.2. Knowledge of Definition of Gender-Based Violence among HCWs

Table 2 describes the responses of the participants’ ability of HCW to define gender-based violence. This was based on a single open-ended question “what is gender-based violence?”. The response was scored on a three-point scale (good, moderate, and poor). This was based on the WHO definition of gender-based violence [27]. According to WHO, the correct definition captures the range of acts by the perpetrator, subjective experiences of the victim, consequences of harm, and form of violence. A respondent that mentioned all items was scored as good; if a respondent mentioned any of the two aspects, it was scored moderate, and if none was mentioned or the respondent cited, they did not know it was scored poorly. Out of 202 participants, 71.79% of HCWs had moderate to good knowledge of GBV.

Additionally, qualitative findings depicted that HCWs understand gender-based violence.

“*Gender-based violence is any act or violation of human right to an individual particularly a woman or girls without those results in injury or harm. It is mostly in form of sexual and physical violence*.” #HCW 2

### 3.3. Knowledge of GBV Management Guideline for Health Sector Response for Prevention of GBV among HCWs

Table 3 describes information about knowledge of gender-based violence management guidelines among HCW. Knowledge of GBV management guidelines was measured on a five-point Likert-type scale (strongly disagree, disagree, neutral, agree, and strongly agree). For statistical analysis purposes, strongly disagree and disagree knowledge was considered to be one category, and also agree and strongly agree knowledge was considered to be one category. Out of all 202 HCWs, only 151 had seen and used a GBV management guideline. After grading the responses, in general, only 36.9% (*n* = 56) of the participants illustrated that they had a good or very good degree of knowledge about the national management guideline for the health sector response to and prevention of GBV, most of them (*n* = 63, 41.6%) illustrated that their knowledge was very poor/poor. In addition, the mean value for all the items was less than 3, which indicates poor knowledge of the management guidelines. The least knowledgeable theme was the role of receptionist in management of GBV (*n* = 27, 17.9%). One of the most knowledgeable theme, as illustrated by the respondents, was GBV management guideline offers a comprehensive framework to guide HCWs in the management of GBV survivors (*n* = 83, 55%). Additionally, majority (*n* = 87, 57.6%) mentioned that GBV guideline address violence against adult men. This is contrary to the guideline, which only addresses violence against children, adolescents, and women.

To expound further on this, qualitative findings revealed that knowledge of GBV management guidelines was helpful in the management of GBV; however, most HCWs emphasized that they were not knowledgeable enough on the guideline, and even some did not even have the guideline at the health facilities.

“*GBV management guidelines are definitely very helpful but some of us have not been trained on GBV management and so do not possess the required knowledge to manage GBV*” #HCW 9

“*I know of the GBV management guidelines but personally I do not know its details and how to use it. Furthermore, we do not have it at our facility. When have a case we refer them to the GBV focal person*” #HCW 11

Some HCWs emphasized very good knowledge of GBV management guidelines as outlined by the process patients take when they access GBV health services.

“*We triage and categorize cases. Patients who come directly to the facility due to violence are treated as a priority and given immediate attention. These cases are handled directly by the GBV focal person and the doctor and sometimes if evidence is required, they involve the laboratory officer. Most patients come from police stations with PF3. Some cases are referred from other departments like CTC. Counselling comes first, we have to talk to and listen to the patient, know how she was violated then after counselling now is when you can manage the patient*.” #HCW 1

### 3.4. Implementation of Guideline for the Health Sector Response to and Prevention of GBV

Out of 202 participants, 60.3% (*n* = 151) were aware of the GBV management guideline for the prevention of GBV. Additionally, 68.2% had a copy of the GBV management guideline. Only 56.9% (*n* = 115) had received any training pertaining to gender-based violence, while only 76.7% of those who had the guideline at the facility received training specific to GBV management guidelines. Only 55.3% of HCWs reported implementing the GBV management guideline in their daily practice with GBV survivors related to the prevention, assessment, diagnosis, and treatment of GBV survivors as illustrated in Table 4 below.

Additionally, HCWs pointed out further training on the use and application of GBV guidelines, filling the police form 3 as part of forensic evidence collection, investigating perpetrator, and more practical sessions.

From the qualitative study, some HCWs reported being aware of the GBV management guideline and incorporating them into their daily practice. Some expressed satisfaction with the wide range of resources that had been supplied to them by the Government to assist with the implementation of the guideline; however, others pointed to lack of resources and lack of the GBV guideline.

“*We thank the government for providing us with necessary kit to assist during forensic investigation*.” #HCW 7

### 3.5. Perceived Gaps in the GBV Management Guideline: HCWs Perspective

Table 5 below describes HCWs’ perspectives on the gaps in the Tanzania GBV management guideline. The majority of HCW mentioned that the guideline lacked a referral director for GBV survivors (79.7%), HCW were not conversant with the language used in the guideline (67.8%), and about 66.8% mentioned that the guideline was not clear when it comes to exemption of GBV health services fee.

Qualitative studies revealed similar results. Some HCWs had the English version of the guideline, which they pointed out to be difficult to understand; instead, they preferred the one written in the national language (Swahili).

“*We received the English version of the GBV management guideline but no everyone is conversant with it so we also request for the Swahili version*.” #HCW 10

The majority of HCWs expressed dissatisfaction with the referral directory.

“*Referral to other specialist services is a problem at the same time there are so many girls who go to the report to the police GBV desk but GBV survivors never visit the health facility*.” #HCW14

## 4. Discussion

This study has highlighted the knowledge of GBV health services and the implementation of GBV health guidelines among HCWs. The majority of HCWs had good knowledge of GBV, contrary to other studies carried out in other African countries and Asian countries [24] that indicated that most health worker were unaware of important aspects of epidemiology of violence, and just over half of the workers had adequate knowledge about gender-based violence.

However, knowledge specific to GBV management guidelines for health sector response to the prevention of GBV among HCW was not adequate. The deficient knowledge of GBV management guidelines among HCWs may be due to a lack of investment in staff training specific to GBV management guidelines. Some of the results of our study have been reported in other studies, such as the lack of GBV knowledge and training to recognize abuse [28]. The danger of this is treating only the immediate complaint and missing the opportunity to provide more comprehensive care to GBV survivors. This was also evident since most HCWs indicated that the training they received was not specific to GBV management guidelines. However, other studies reported adequate knowledge of both PMTCT and GBV guidelines [29,30]. This discrepancy might be due to a difference in knowledge in guidelines, methodology, sample size, and sociodemographic differences.

The majority of aspects that HCWs pointed out for further training focused on some of the components of the GBV guideline. Training should go beyond the GBV guidelines to help the health provider understand the patient’s cultural and religious beliefs. This was similar to other studies conducted among midwives; they believed that training in GBV and GBV guidelines would increase the rate of detection of violence in maternity settings, thereby presenting an opportunity for women to access prevention services for GBV early [31]. Interestingly, some cultural and social norms may perpetrate specific forms of violence. A study conducted in Tanzania reported that AGYW expressed that men have a right to control or discipline women through physical means, and women’s acceptance of this made them more vulnerable to continued violence [32]. This purports that cultural and social norms value men as superior and more powerful than women, and these norms and cultures subordinate women in many life spheres, from economic independence to decision-making power. This often happens in contexts where societal norms allow the use of GBV to reprimand women and where men are expected to have the final say as a means to control women [33]. These observations are similar to studies conducted in Nigeria [34,35] where women described that cultural norms encouraged them to tolerate and accept acts of violence perpetrated against them hence preventing them from opening up during their encounters with HCWs. These prevailing cultural gender norms may additionally give insight as to why physical and sexual violence in intimate relationships (e.g., domestic rape) is still considered culturally acceptable in Tanzania [36]. Additionally, marital rape is not recognized by Tanzanian law hence preventing married women from seeking help and obtaining appropriate services following sexual violence [37].

Regarding the implementation of GBV management guidelines by HCWs, our study revealed that very few health workers used the guideline while managing GBV cases. A similar assessment of HIV, STI, and GBV screening and treatment practices has been reported by Wyndham-Thomas et al. [38]. The study reported marked inconsistencies in screening and treatment approaches among patients attributing these inconsistencies to the absence of up-to-date guidelines. Similarly, a study in Malawi revealed that the Ministry of Health collaborated with the Police Service to develop, disseminate and promote the implementation of the GBV management guideline; however, the majority of clinicians were unaware of the existence or content of the national guidelines [39]. In recognition of this issue, the Ministry of Health should train HCWs on the procedures outlined in the GBV management guideline to strengthen health services. The need to train HCWs as a way of strengthening the health system’s response to violence against women in Uganda was also highlighted by the WHO report on lessons learned from adapting and implementing WHO guidelines and tools [40].

The availability of a management guideline indicates a positive sign toward the management of GBV; however, without an appropriately communicated guideline, care providers, patients, and communities are not guided adequately on the best practices to yield better outcomes [41]. Identifying gaps in any management guideline is the process of designing and fitting the recommendations to the implementation environment and vice versa. HCWs in our study identified that the language used within the GBV management guideline was difficult to follow and use. Additionally, there was not enough clarification on the exemption of GBV health care, lack of information on the management of sexual violence against children, and referral protocol.

It is very important to take prompt action in order for the GBV survivor to access quality and timely care. In addition, a report by the Government of Tanzania highlighted the need to improve training for HCWs, integrate GBV screening and comprehensive care into existing structures, and strengthen coordination among service providers and referral systems as a way to improve the quality of care for GBV survivors [22]. Timely referrals can save lives and prevent further harm and medical consequences in some of the violence, especially sexual violence [42]. However, our study participants revealed that there was no clear referral protocol to assist GBV survivors. It is paramount to develop a clear system of referral protocol with referral focal points at every health facility.

## 5. Conclusions

The Government of Tanzania is in the process of developing a revised National Plan of Action and the GBV management guideline (NPA-VAWC) for both Mainland and Zanzibar for the next 5 years. This is an important tool that highlights how well-trained HCWs can identify and provide a supportive response in making an enormous difference in the lives of women and reducing the impact and recurrence of violence. The government has tried to distribute the guidelines to almost all health facilities; however, HCWs in our study reported not using the guideline during the management of GBV. In addition, HCWs reported that the guideline lacked a referral directory, had a difficult language, and was written in English which is also difficult to comprehend as it is not the national language. The next management guideline should be translated into Kiswahili and revised in a manner that can be easily comprehended. In addition, there is a need to build the capacity of health workers to implementation of GBV guidelines while providing GBV management either through continuous health education programs or through regular job practical training as well as regular workshops or seminars involving practical sessions. This is one of the most long-lasting and effective measures to improve health workers’ knowledge and foster compliance with the implementation of GBV management guidelines.

## Figures and Tables

**Figure 1 ijerph-20-05409-f001:**
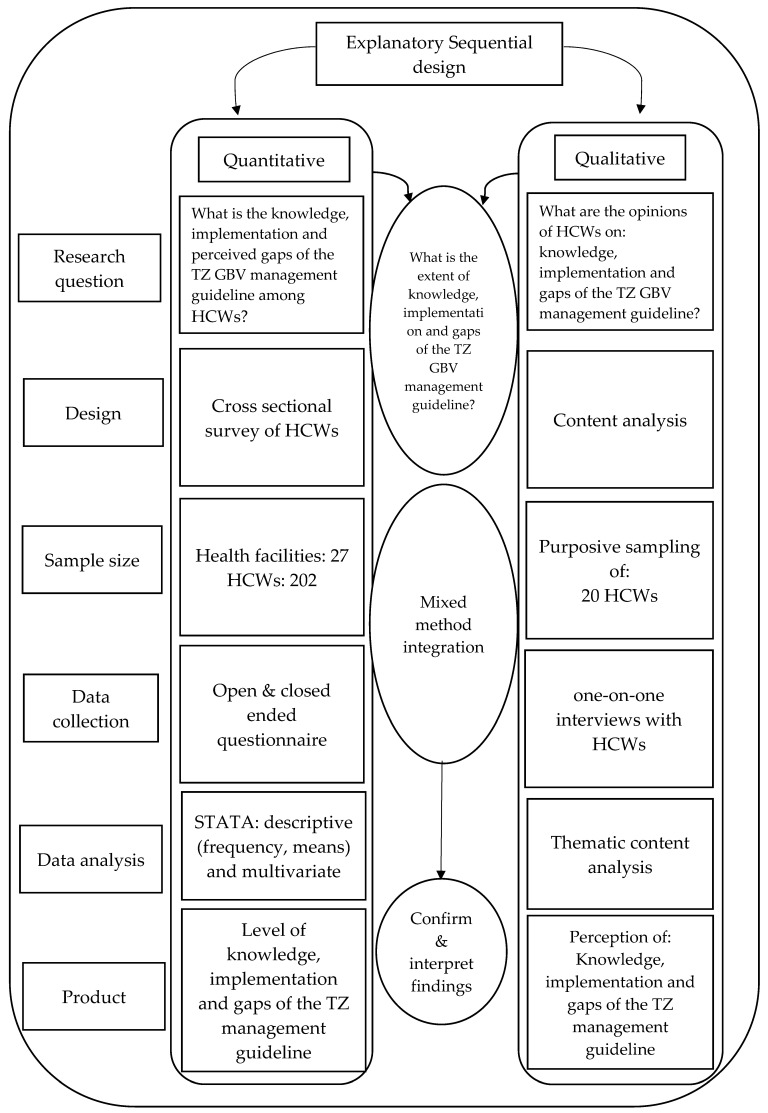
Explanatory sequential mixed methods procedural diagram.

**Table 1 ijerph-20-05409-t001:** Sociodemographic characteristics.

Characteristics	Frequency *n* (%)
District name	
Kinondoni	138 (68.3)
Temeke	64 (31.7)
Age	
23–33	57 (28.2)
34–44	88 (43.6)
45–55	43 (21.3)
56>	14 (6.9)
Gender	
Male	58 (28.7)
Female	144 (71.3)
Level of education	
undergraduate education	17 (8.4)
Diploma/certificate	174 (86.4)
Master’s education	11 (5.5)
Health facility level	
Dispensary	64 (31.9)
Health center	122 (60.4)
Hospital	16 (7.9)
Health cadre	
Clinical officer	52 (25.7)
Specialist doctor	4 (2.0)
Medical officer	9 (4.5)
Nurses/midwife	122 (60.4)
Assistant medical officer	15 (7.4)
Facility department	
Outpatient department	127 (31.5)
Reproductive and child health clinic	97 (24.1)
Counseling and treatment clinic	29 (7.2)

**Table 2 ijerph-20-05409-t002:** Knowledge of gender-based violence (GBV).

	Degree of Knowledge (*n* = 202)		
	Good	Moderate	Poor
Ability to define GBV	104 (51.49%)	41 (20.30%)	57 (28.22)

**Table 3 ijerph-20-05409-t003:** Knowledge of gender-based violence (GBV) management guidelines among HCW.

Variables	Degree of Knowledge (*n* = 151)			
	Strongly Agree/Agree	Neutral	Very Disagree/Disagree	Mean
GBV management guideline offers comprehensive framework for management of GBV survivors	83 (55.0%)	31 (20.5%)	37 (24.5%)	1.17
Survivors have a right to request a female or male health workers	41 (27.2%)	63 (41.7%)	47 (31.1%)	1.53
GBV guideline addresses economic violence	36 (23.8%)	34 (22.5%)	81 (53.6%)	0.92
GBV guideline addresses psychological violence	59 (39.1%)	44 (29.1%)	48 (31.8%)	1.27
GBV guideline address children, adolescents, and women violence	78 (51.7%)	41 (27.2%)	32 (21.2%)	1.34
GBV guidelines address violence against adult men	87 (57.6%)	36 (23.8%)	28 (18.5%)	1.30
GBV guidelines explain on linkage between GBV prevention and services for survivors	38 (25.2%)	18 (11.9%)	95 (62.9%)	0.61
GBV guidelines address need for HCW to testify in court on behalf of the survivor	82 (54.3%)	22 (14.6%)	47 (31.1%)	0.99
GBV guideline states that it is the HCW responsibility to collect forensic evidence	49 (32.5%)	21 (13.9%)	81 (53.6%)	0.75
GBV guideline states role of Pharmacist in management of violence	39 (25.8%)	14 (9.3%)	98 (64.9%)	0.54
GBV guideline states role of Laboratory Technician in management of violence	69 (45.7%)	29 (19.2%)	53 (35.1%)	1.04
GBV guideline states role of Receptionist in management of violence	27 (17.9%)	32 (21.2%)	92 (60.9%)	0.82
GBV guideline indicates minimum standards of GBV management by level of health facility	43 (28.4%)	28 (18.5%)	80 (53.0%)	0.84
GBV guideline states that when obtaining consent, the survivor should be briefed on GBV-related health consequences	47 (31.1%)	56 (37.1%)	48 (31.8%)	1.43
GBV guideline states that clinicians should fill out the PF3 form	77 (51.0%)	23 (15.2%)	51 (33.8%)	0.97
GBV guideline states that children have a role in decision making	39 (25.8%)	27 (17.9%)	85 (56.3%)	0.80
GBV guideline indicates that all GBV treatment should be given free of charge	52 (34.4%)	35 (23.2%)	64 (42.4%)	1.04

Values are expressed as *n* (%).

**Table 4 ijerph-20-05409-t004:** Questions regarding implementation.

Questions Regarding Implementation	Yes (%)
Did you receive training pertaining to GBV? (*n* = 202)	115 (56.9)
Are you aware of the GBV management guideline? (*n* = 202)	151 (60.3)
- Do you have guidelines for managing gender-based violence? (*n* = 151)	103 (68.2)
- Did you receive training pertaining to GBV management guidelines? (*n* = 103)	79 (76.7)
- Do you use the guideline as a guide for management of GBV? (*n* = 103)	57 (55.3)
- Is the GBV guideline practical and useful? (*n* = 57)	21 (36.8)

Values are expressed as *n* (%).

**Table 5 ijerph-20-05409-t005:** Gaps of gender-based violence (GBV) management guidelines.

Themes (*n* = 202)	Yes	No
Language not easily understood	137 (67.8%)	65 (32.2%)
Some treatments mentioned are not available in the guideline; hence, guidelines are not relevant	107 (53.0%)	95 (47.0%)
Following the guidelines is difficult	102 (50.5%)	100 (49.5%)
Lack of clear clarification on GBV health care fee exemption	135 (66.8%)	67 (33.2%)
Lack of referral directory	161 (79.7%)	41 (20.3%)
Lack of enough information on management of sexual violence against children <10 years	109 (54.0%)	93 (46.0%)

Values are expressed as *n* (%).

## Data Availability

The full dataset (transcripts) generated and analyzed during the current study is not publicly available because they contain information that could compromise the privacy of research participants. They can be made available on request from the corresponding author due to privacy concerns of the nature of the study but can be made available from the corresponding author (C.M.).

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
