# Peer review of "Knowledge, Implementation, and Gaps of Gender-Based Violence Management Guidelines among Health Care Workers"

_ijerph, 2023, doi:10.3390/ijerph20075409_

Round 1

Reviewer 1 Report

First of all, thank you for giving me the opportunity to review this article which deals with a crucial issue for social progress. The researchers did a great deal of fieldwork and collected a lot of qualitative information that was reported in the paper. It cannot be said that a representative sample was obtained but the topic dealt with, the difficulty in dealing with it and the importance of proposing studies on GBV certainly make it acceptable.
However, some weaknesses in the exposition require refinement of the text of the paper. First of all, it is necessary to improve the reference to the literature and to cite the papers with greater precision as it seems that often the citations are rather generic and do not correspond much to the focus of the cited work. Furthermore, there is much talk of recommendations from international bodies, it would be advisable to report these recommendations or at least insert the link to the websites that report them.
Finally, a consideration on the quantitative data presented: it seems that even those who replied that they have never heard of the guidelines (Are you aware of the GBV management guideline?
164 (81.2)) was then asked to judge its contents in terms of comprehensibility and practicability. How could those who have never seen them have answered with knowledge of the facts? Perhaps the presentation of quantitative data should be reviewed using appropriate filters that refer the answers to only the part of respondents who may have given a coherent opinion.

Author Response

Kindly find responses to your comments in the attachment

Reviewer 2 Report

This is a well-written paper that examines the knowledge and implementation of gender based violence (GBV) management guidelines among healthcare workers in Tanzania and also assesses the gaps in knowledge and implementation. The topic is important for public health research as healthcare workers are first responders and gaps in knowledge, as well as in implementation of GBV management guidelines, compromises the quality of care provided. The methodology is sound and the findings are relevant for research in this area. 

I feel the discussion section can be elaborated further. The authors have provided some circumspection on the reasons behind deficiency in knowledge of GBV management guidelines among Tanzanian healthcare workers but a more detailed discussion will strengthen the study. The authors note that "Training should go beyond the GBV guidelines to help the health provider understand the patient’s cultural and religious beliefs" (page 9). Perhaps a short analysis on the impact of cultural and religious contextualization of attitudes towards GBV can address gaps in knowledge, and overall enhance the quality of GBV training guidelines. Also, please make the formatting of the tables consistent throughout the paper.  

Author Response

(The authors gave the same response as above.)

Round 2

Reviewer 1 Report

Thank you for accepting my suggestions and improving the comprehension of this paper which remains a very important attempt to bring to the attention of everyone, both the scientific community and public opinion and politics, a topic of great importance.